# *POLD1* DEDD Motif Mutation Confers Hypermutation in Endometrial Cancer and Durable Response to Pembrolizumab

**DOI:** 10.3390/cancers15235674

**Published:** 2023-11-30

**Authors:** Christina Hsiao Wei, Edward Wenge Wang, Lingzi Ma, Yajing Zhou, Li Zheng, Heather Hampel, Susan Shehayeb, Stephen Lee, Joshua Cohen, Adrian Kohut, Fang Fan, Steven Rosen, Xiwei Wu, Binghui Shen, Yuqi Zhao

**Affiliations:** 1Department of Pathology, City of Hope Medical Center (COHNMC), Duarte, CA 91010, USA; ffan@coh.org; 2Department of Oncology & Therapeutics Research, City of Hope Medical Center (COHNMC), Duarte, CA 91010, USA; edwang@coh.org; 3Department of Cancer Genetics and Epigenetics, Beckman Research Institute at City of Hope Medical Center (COHNMC), Duarte, CA 91010, USA; lingzima@stanford.edu (L.M.); yajingzhou@ujs.edu.cn (Y.Z.); lzheng@coh.org (L.Z.); bshen@coh.org (B.S.); 4Clinical Cancer Genetics, City of Hope Medical Center (COHMC), Duarte, CA 91010, USA; hhampel@coh.org (H.H.); sshehayeb@coh.org (S.S.); 5Division of Gynecologic Oncology and Surgery, City of Hope Medical Center (COHNMC), Duarte, CA 91010, USA; stelee@coh.org (S.L.); jgcohen@coh.org (J.C.); akohut@usf.edu (A.K.); 6Department of Hematology & Hematopoietic Cell Transplantation, City of Hope Medical Center (COHNMC), Duarte, CA 91010, USA; srosen@coh.org; 7Beckman Research Institute of City of Hope, Duarte, CA 91010, USA; xwu@coh.org (X.W.); yuqzhao@coh.org (Y.Z.)

**Keywords:** DNA polymerase delta, POLD1, endometrial cancer, hypermutation, germline, variant of uncertain significance, immunotherapy

## Abstract

**Simple Summary:**

Germline mutations in the DNA polymerase delta 1 (POLD1) exonuclease domain cause DNA proofreading defects, tumor hypermutation, and predispose to hereditary colorectal and endometrial cancer. In this work, we used a multiprong approach to demonstrate that mutations that alter the amino acid charges in the DEDD motif of exonuclease domains could lead to proofreading deficiency. This novel mutation class, which is currently classified as a variant of uncertain significance, should be reclassified as likely pathogenic.

**Abstract:**

Background: Mutations in the DNA polymerase delta 1 (POLD1) exonuclease domain cause DNA proofreading defects, hypermutation, hereditary colorectal and endometrial cancer, and are predictive of immunotherapy response. Exonuclease activity is carried out by two magnesium cations, bound to four highly conserved, negatively charged amino acids (AA) consisting of aspartic acid at amino acid position 316 (p.D316), glutamic acid at position 318 (p.E318), p.D402, and p.D515 (termed DEDD motif). Germline polymorphisms resulting in charge-discordant AA substitutions in the DEDD motif are classified as variants of uncertain significance (VUSs) by laboratories and thus would be considered clinically inactionable. We hypothesize this mutation class is clinically pathogenic. Methods: A review of clinical presentation was performed in our index patient with a POLD1(p.D402N) heterozygous proband with endometrial cancer. Implications of this mutation class were evaluated by a Preferred Reporting Items for Systematic Reviews and Meta-Analyses (PRISMA)-guided systematic review, in silico analysis with orthogonal biochemical confirmation, and whole-exome and RNA sequencing analysis of the patient’s tumor and engineered cell lines. Results: Our systematic review favored a Mendelian disease mutation class associated with endometrial and colorectal cancers. In silico analysis predicted defective protein function, confirmed by biochemical assay demonstrating loss of nuclease activity. A *POLD1*-specific mutational signature was found in both the patient’s tumor and *POLD1*(p.D402N) overexpressing cell. Furthermore, paired whole-exome/transcriptome analysis of endometrial tumor demonstrated hypermutation and T cell-inflamed gene expression profile (GEP), which are joint predictive biomarkers for pembrolizumab. Our patient showed a deep, durable response to immune checkpoint inhibitor (ICI). Conclusion: Charge-discordant AA substitution in the DEDD motif of *POLD1* is detrimental to DNA proofreading and should be reclassified as likely pathogenic and possibly predictive of ICI sensitivity.

## 1. Introduction

The *POLD1* gene encodes the 125 kDa catalytic subunit of the heterotetrameric POLδ complex. *POLD1* catalyzes two essential functions of POLδ: synthesizing Okazaki fragments with its DNA polymerase activity and DNA proofreading with its 3′ to 5′ exonuclease activity [1]. In humans, germline mutations in *POLD1* resulting in defective DNA proofreading exonuclease function are associated with hereditary colorectal and endometrial cancer and colonic polyposis. Clinically, these tumors a exhibit high tumor mutation burden (TMB), increased presence of tumor infiltrating lymphocytes (TIL), and favorable response to immune checkpoint inhibitors [2]. Therefore, accurate detection of all affected patients has the potential to significantly impact their clinical management, including cancer surveillance, genetic counseling of family members, and precision oncology for immunotherapy.

*POLD1* contains several important conserved motifs: Amino acid residues 4–19 form the nuclear localization signal, 304–533 form the exonuclease domain, and 579–974 form the polymerase domains [3]. *POLD1*’s exonuclease domain is a member of the DEDD-type exonuclease (Exo) superfamily characterized by three sequence motifs: Exo1, Exo2, and Exo3 [3]. Within the three sequence motifs, there are four highly conserved, invariant catalytic amino acids, DEDD, which have carboxylic sidechains. The DEDD motif is composed of p.D316 (located in Exo1), pE318 (located in Exo1), p.D402 (located in Exo2), and p.D515 (located in Exo3) (Figure 1A). The DEDD amino acids use their negatively charged carboxylate sidechains to coordinate the positively charged divalent metal ions to catalyze a phosphoryl-transfer reaction, thereby removing misincorporated nucleotides [4,5]. In yeast, mice, and human normal intestinal cells, *POLD1*’s DNA proofreading function is disrupted when an amino acid in the DEDD motif is mutated with a non-acidic amino acid lacking a carboxylate sidechain (charge-discordant AA substitution) [6,7,8,9].

Currently, virtually all clinically actionable pathogenic/likely pathogenic *POLD1* variants (p.S478N and p.L474P) are missense mutations within the exonuclease DNA proofreading domain and spare the DEDD motif (Figure 1A). Our search of the ClinVar database [10] for germline missense mutations causing charge-discordant AA substitution in the DEDD motif yielded thirteen variant entries: all are classified as variants of uncertain significance (VUS; Appendix A). This designation is clinically inactionable, and additional supporting evidence is required for their reclassification. Due to the mechanistic significance of these four carboxylic acid sidechains in the DEDD motif, we hypothesize charge-discordant AA substitutions would result in defective exonuclease function, DNA proofreading errors, and be tumorigenic in humans.

Herein, we studied an index case of a young woman with early onset endometrial cancer, who was found to be a carrier of a charge-discordant AA substitution in the DEDD motif (p.D402N). Using a multiprong approach, we systematically investigated the pathogenic effect of this mutation class. Our goal is to provide evidence to support its reclassification to pathogenic/likely pathogenic, so that these patients will obtain appropriate genetic counseling and cancer screening. Furthermore, recent data have shown that *POLE* and *POLD1* mutation status is a predictive biomarker for immune checkpoint inhibitor response. Failure to reclassify these variants could lead to opportunity loss for personalized immune checkpoint inhibitor therapy.

## 2. Materials and Methods

### 2.1. Ethics Statement and Institutional Study Approval

Patient consent was obtained to participate in the study, which was approved by the COH review board IRB#07047, IRB#22078, and IBC#22012.

### 2.2. Systematic Analysis of Genotypic-Phenotypic Association of Germline POLD1 Mutations Involving the DEDD Motif

To identify relevant papers, a PUBMED search was performed for all original papers published up to 15 May 2023 using the following keyword search terms: (i) “*POLD1* and germline” and (ii) “DNA polymerase and germline”. We did not apply language restrictions. The search yielded 310 hits, and their full-length papers were manually screened for relevance. Studies were excluded based on the following exclusion criteria: (i) no germline information (somatic only) or no genotype reporting and (ii) no reporting of alterations in the DEDD motif (p.D316, p.E318, p.D402, p.D515). Duplicated reporting of the same patient in different journals was recorded together to avoid cout duplication. The PRISMA 2020 flow diagram is presented in Appendix A.

### 2.3. In Silico Analysis Predicting Functional Effect of DEDD Domain Alteration

To assess the functional impact of all 13 variants with nonacidic amino acid changes to the DEDD motif, we applied six different computational algorithms to test each variant. The algorithms used include PolyPhen-2, MutationTaster, Panther, Mutation Assessor, SIFT, and MutPred2SIFT [11,12,13,14,15,16]. The in silico online algorithms were accessed between January 2022 to January 2023 (Appendix A for website addresses). 

### 2.4. Human POLδ Expression, Purification, and In Vitro Polymerase and Exonuclease Assay

POLDδ is composed of 4 subunits, POLD1 (p125), POLD2 (p68), POLD3 (p50), and POLD4 (p12). Recombinant WT or p.D402A POLD1, POLD2, POLD3, or POLD4 was expressed in the SF-9 insect cell line. They are purified and reconstituted according to previously published protocols with modifications [17,18]. Briefly, the cells were pelleted using 500 mL of SF-9 infected cells (multiplicities of infection = 1, 72 h), then sonicated using a 40 mL lysis buffer (4 × 15 s bursts). The supernatant (40 mL) and 78F5 beads (10 mL) were incubated overnight and subsequently loaded on the Mono Q column (at increments of 10 mL beads), washed using a 10 bed volume of TEGG/0.4M NaCl buffer (consisting of Tris-HCL (pH 7.8), 10% glycerol, 0.5 mM ethylene glycol-bis(β-aminoethyl ether)-N,N,N′,N′-tetraacetic acid (EGTA), 1 mM ethylenediaminetetraacetic acid (EDTA), and 0.4 M NaCl), and eluted using a 5 bed volume of TEGG/0.4 M NaCl plus 30% ethylene glycol. POLδ subunits were verified on a sodium dodecyl sulfate-polyacrylamide gel electrophoresis (SDS-PAGE) showing appropriate banding patterns. Purified WT or p.D402A POLD1 was reconstituted with POLD2, POLD3, and POLD4 to make WT or D402A POLδ in vitro. To assay the polymerase or 3′ exonuclease activity of WT or p.D402A POLδ, 5 µM FAM-labeled gapped DNA duplex was incubated with 50 ng POLδ (WT or D402A) and 50 ng proliferating cell nuclear antigen (PCNA) in a reaction buffer (20 mM Tris-HCL, pH7.5, 8 mM MgOAc2, 1 mM dithiothreitol (DTT), 0.1 mg/mL bovine serum albumin (BSA) with or without 50 nM four deoxyribonucleotides (dNTPs). The reaction was carried out at 37 °C for 30 min and resolved on 15% denaturing PAGE. 

### 2.5. Generating Endometrial Cell Line Overexpressing POLD1 p.D402N

The HEC1A cell line was purchased from ATCC, and maintained in Dulbecco’s Modified Eagle Medium-high glucose (DMEM-H) supplemented with 10% fetal bovine serum (FBS) and sub-cultured every 2–3 days at 1:3–1:5 ratios. Sterility tests for bacteria, fungi, and mycoplasma were negative. To generate a HEC1A stable cell line that overexpresses either wild-type (wt) and h*POLD1* p.D402N variants, the HEC1A cells were infected with lentiviral construct pLV[Exp]-Puro-CMV>h*POLD1*[NM_002691.4] and pLV[Exp]-Puro-CMV>h*POLD1*[NM_002691.4] (D402N), respectively, as previously described [19]. After transduction, cells were maintained in 3.0 µg/mL puromycin containing growth media. The vector design, construction, virus packaging, cell transduction, and generation of pooled stable cell lines were performed by VectorBuilder (Chicago, IL, USA). *POLD1* overexpression was confirmed by qPCR. The genotype of the generated stable cell line was confirmed using whole-exome sequencing, which confirmed the presence of h*POLD1* p.D402N in the cell lines infected with the lentiviral construct containing this variant. Baseline *PMS2* mutation in HEC1A cell lines was also verified using WES. 

### 2.6. Whole-Exome Sequencing (WES)

For cell line experiments, the WES experiments were performed in three biological replicates for each group to identify the somatic mutations among them. For library preparation, the qualified genomic DNA was randomly broken into 180–280 bp fragments. Exonic DNA capture was conducted with Agilent SureSelect Human All ExonV6 (Agilent Technologies, Santa Clara, CA, USA) following the manufacturer’s instructions and was subsequently sequenced using the Illumina NovaSeq 6000 platform for paired-end reads of 101 bp. WES was performed on wild type and mutant groups at passage p4 and p20 at mean coverages of 115X. For patient tumors, the formalin-fixed, paraffin embedded (FFPE) tumor tissue was used as source of DNA for WES. The DNA extraction, library preparation, and sequencing methods have been previously described, and were performed in a CLIA-certified laboratory [20]. Briefly, the nucleic acid is extracted using the Qiagen AllPrep DNA/RNA FFPE kit. The DNA quality control metric was confirmed with 260/280 measurement. The DNA library was prepared using KAPA HyperPre library kit (Roche), and the process includes end repair and A-tailing, followed by library amplification (~300 bp length, a minimum of 500 ng yield). Targeted sequences from the DNA library were captured using a custom exome capture probe set. The sequencing was performed at a CLIA-accredited laboratory that performs clinical-grade sequencing (Illumina NovSeq 6000 platform) for clinical applications. The FASTQ data was transferred to COHNMC’s Center for Informatics and released to the investigators through a data honest broker process. 

### 2.7. Whole-Transcriptome Sequencing

Whole-transcriptome sequencing was performed on the patient’s endometrial FFPE tumor sample, the source of RNA for whole-transcriptome sequencing. The RNA extraction, library preparation, and sequencing methods have been previously described [20]. The sequencing run was performed in a CLIA-certified laboratory [20]. The RNA-seq data was transferred to COHNMC’s Center for Informatics and released to the investigators through a data honest broker process. 

### 2.8. Somatic Mutation Calling

High-quality reads were obtained by trimming the raw reads using fastp (v0.23.3) [21] and by carrying out FastQC (v0.11.9) for the quality control assessment. The clean reads were aligned to the human genome GRCh38.p13 with Burrows–Wheeler alignment (v0.7.17) [22]. Then, Genome Analysis Toolkit (GATK, v4.1.8.0) [23] was used for variant calling using the best practices workflow. The raw somatic mutation data was normalized and filtered using bcftools (v1.9) [24] with the sequencing read depth > 10, the filtering criteria satisfying “PASS”, and alternate allele frequency > 0.05. The analysis-ready variants were annotated and converted to the mutation annotation format (MAF), and the R package maftools (v2.16.0) [25] was applied to screen the mutated genes in each group. 

### 2.9. T Cell-Inflamed GEP Score Calculation

The RNA-seq data were analyzed using fastp-HISAT2-STRINGTIE2. Then, read count was normalized by library size to obtain counts-per-million (CPM) reads, which was then log transformed. The T cell-inflamed GEP score of a sample was calculated as the ssGSEA normalized enrichment score of the 18 inflammatory genes (*CCL5*, *CD27*, *CD274* [PD-L1], *CD276* [B7-H3], *CD8A*, *CMKLR1*, *CXCL9*, *CXCR6*, *HLA-DQA1*, *HLA-DRB1*, *HLA-E*, *IDO1*, *LAG3*, *NKG7*, *PDCD1LG2* [PD-L2], *PSMB10*, *STAT1*, and *TIGIT*), as previously described [26].

## 3. Results

### 3.1. Case Control Study

Our index patient is a 32-year-old woman who carries a germline heterozygous *POLD1* p.D402N (c.1204G>A) variant. This alteration converts the negatively charged aspartic acid to the neutrally charged asparagine lacking carboxylate sidechain at AA position 402 in the DEDD motif. She was diagnosed with stage IVB International Federation of Gynecology and Obstetrics (FIGO) grade 3 endometrioid adenocarcinoma with a significant amount of residual tumor following neoadjuvant therapy, including metastasis to the para-aortic lymph nodes, renal artery, and umbilicus. MLH1, PMS2, MSH2, and MSH6 were all expressed in tumor cells by immunohistochemistry, lending no support for mismatch-repair deficiency. Clinical-grade whole-exome sequencing (WES) of the endometrial tumor revealed high TMB (141 mut/Mb), microsatellite stable status, and loss of heterozygosity for the *POLD1* allele. Microscopic examination of her tumor tissue showed abundant tumor infiltrating lymphocytes (Figure 1B), which corroborate her high tumor mutation burden status, suggestive of increased neoantigen production. For adjuvant therapy, she received Pembrolizumab every two weeks (>29 cycles) and Lenvatinib daily. She showed a deep and durable response to immune check point inhibitor therapy: Thirty months later, she remains cancer-free.

Her family history showed endometrial cancer in two maternal aunts and her paternal grandmother (Figure 1C). The large population ExAc dataset (gnomAD) [27] (https://gnomad.broadinstitute.org/variant/19-50403559-G-A?dataset=gnomad_r4 accessed on 29 November 2023) revealed ultralow allele frequency (1.3 × 10^−6^) for this variant. Taken together, the phenotype suggests the possibility that this *POLD1* variant may contribute to the development of endometrial cancer in this family.

### 3.2. The Genotypic–Phenotypic Association of Germline POLD1 Mutations Involving the DEDD Motif from Existing Literature Revealed These Affected Individuals Present with Cancers in Colon, Endometrium, and Breast

To find additional cases to support the genotypic–phenotypic association of germline *POLD1* mutation involving the DEDD motif, we performed a PUBMED search. Our systematic analysis led to the inclusion of 4 peer-reviewed papers in this analysis (PRISMA 2020 flow diagram in Appendix A) [9,28,29,30]. The most frequently reported variant occurred on AA position 316. The reported variants included p.D316G, p.D316H, p.D316N, which all led to the loss of the carboxylic acid sidechain [9,28,29]. Other reported variants, p.E318K and p.D402N were reported in one patient each [30]. We found that charge-discordant AA substitutions in the DEDD motif were detected in patients registered with colorectal cancer consortiums from multiple continents, including Europe and Asia (Table 1). Three variants (p.D316G, p.D316H, p.D316N) were detected in five patients from European consortium registries [9,28]. The phenotypic characteristics of these carriers included colorectal cancer (2 patients; age presentation 44 and 58), endometrial cancer (3 patients; age presentation 54, 54, and 57), and breast cancer (age presentation 64). Of the five patients, one patient presented with both colorectal and endometrial cancer (diagnosed at 44 and 54 years old, respectively); another patient presented with both endometrial (diagnosed at 57 years old) and breast cancer (age of diagnosis uncertain). In the Asian/Taiwanese cohort, the p.D402N and p.E318K variants were reported, and both patients had colorectal cancer (age and gender not reported). Endometrial cancer was observed in our index patient (diagnosed at 32) and her maternal aunts. On balance, colorectal, endometrial, and breast cancers were found in patients with germline *POLD1* polymorphisms causing discordant AA substitution in the DEDD motif. At the time of the preparation of this manuscript (October 2023), these published variants (p.D316G, p.D316H, and p.D316N) were still classified as VUSs based on the ClinVar database [10].

### 3.3. In Silico Analysis Predicted Deleterious Functional Effect of Charge-Discordant AA Substitutions in the DEDD Domain

To study the functional consequence of charge-altering AA changes in the DEDD motif of POLD1, we applied six computational analyses [11,12,13,14,15,16] to all thirteen POLD1 variants with this mutation class identified from ClinVar database: the result uniformly supported deleterious functional impact (Table 2). 

Substituting the carboxylic acid side chain containing an amino acid with a neutral side chain containing an amino acid in the DEDD motif causes defective exonuclease activity in vitro: to orthogonally validate the results of our in silico functional prediction analysis, we performed an in vitro biochemical assay to assess if such an alteration would lead to a loss of function in either exonuclease or DNA polymerase functions, or both. For this experiment, we opted to alter the negatively charged aspartic acid at DEDD motif AA position 402 to neutrally charged alanine, which represents the D402G/N/V mutants observed in the human population, as a proof of principle that this mutation class shows generalizable functional consequences. First, we expressed and purified WT POLD1 or p.D402A POLD1 proteins. WT POLδ or p.D402A POLδ was reconstituted using the WT POLD1 or p.D4024A POLD1 with POLD2, POLD3, and POLD4 proteins. We assayed the polymerase and 3′-exonuclease activities using fluorescein (FAM)-labeled gap DNA substrates. p.D402A POLDδ had similar polymerase activity to the WT, but it had little 3′ exonuclease activity (Figure 2), supporting our hypothesis that the DEDD domain is crucial for 3′ exonuclease activity. 

### 3.4. Whole-Exosome Sequencing of Endometrial Tumors with POLD1 p.D402N Mutation Revealed POLD1 DNA Proofreading Deficiency-Specific Mutational Signature and a Pan-Tumor Genomic Biomarker for Immunotherapy

If *POLD1* p.D402N is functionally deleterious, we hypothesized that it would leave a genomic footprint characterized by *POLD1* DNA proofreading deficiency mutational signature [9,31]. This mutational signature is characterized by the prominence of C>A substitution at ACC, CCA, CCT, TCA, TCT, TCA, and TCT trinucleotide of the 96 possible substitution mutational repertoire (mutated bases are underlined) [9,31]. Other *POLD1* exonuclease deficiency-associated somatic mutations include increased single base substitution and insertion/deletions (INDELs) [1,9]. In our index patient, we evaluated her endometrial tumor’s whole-exome genome for a *POLD1* DNA proofreading deficiency mutational signature. Of note, the tumor is microsatellite stable. In this context, the mutational profiling analysis revealed a contribution from the SBS10d mutational signature, which is etiologically associated with the *POLD1* proofreading deficiency [31,32]. The loss of the heterozygosity of the *POLD1* gene was also detected. The observed mutational signature pattern in our patient tumor tissue is in keeping with the expected loss of exonuclease activity of *POLD1* and supports our hypothesis that p.D402N negatively impacts the DNA proofreading function. We further hypothesized that the deep, durable anti-tumor response to the immune checkpoint blockade observed in our patient is due to neoantigen production and the immune-hot tumor microenvironment. Indeed, this is supported by a recent study demonstrating the predictive utility of joint high TMB and T cell-inflamed gene expression profile (GEP) scores as a robust pan-tumor genomic biomarker for PD-1 checkpoint blockade-based immunotherapy [26]. We analyzed our patient’s whole-exome and whole-transcriptome sequencing data, which, respectively, demonstrated a high TMB and GEP score (TMB score = 141 mut/Mb; GEP score = 0.25). This corroborates the clinical observation that our endometrial cancer patient showed a favorable and durable response to immunotherapy and showed no evidence of disease 2 years following her initial stage 4 diagnosis.

### 3.5. Elucidating the Impact of p.D402N Mutation on Cancer Genome Using Stably Transfected Cell Lines

While the genomic analysis from the patient’s endometrial tumor suggests the pathogenic role of the *POLD1* p.D402N mutation, it is unclear if the observed increased tumor mutation burden and SBS10d mutational signature are directly caused by the patient’s germline p.D402N. To elucidate the direct biological effect of *POLD1* p.D402N on the genome, we generated two lentiviral transduced stable cell lines that overexpress either the *POLD1* p.D402N mutation or wildtype (wt) *POLD1*. We chose this model because a similar in vitro overexpression model had been previously used to successfully study the functional effect of inactivating mutations in the exonuclease and polymerase domains in *POLD1* [33]. In the current study, we used HEC1A, an endometrial cancer cell line, as a physiologically pertinent model system to study *POLD1* p.D402N function. Of note, the HEC1A cell line is *PMS2* mutated, microsatellite unstable, with a high mutation burden, and contains no preexisting *POLD1* or *POLE* exonuclease deficiency-causing mutations. The pooled engineered cell lines were passaged 20 times to allow mutation accumulation. Bioinformatic analysis endorsed the respective *POLD1* genotypes at passage 20, confirming the stability of the *POLD1* status in our engineered cell lines (Appendix A). Given that the HEC1A cell line has a baseline mismatch repair deficiency (*PMS2* mutation) and high mutation burden, we directed and interpreted our analysis on the unshared mutations between the two comparison groups to be representative of newly gained mutations following lentiviral transduction. At passage 20, *POLD1*(p.D402N) overexpressing cells showed higher global mutation counts, INDELs, and single nucleotide polymorphisms (SNP)/single base substitution, compared to *POLD1*(wt) overexpressing cells (Figure 3A–C). Our result is consistent with prior studies, where these somatic mutations have been observed to accumulate in normal human colonic crypt cells with a germline *POLD1* exonuclease domain mutation [9]. Significantly, *POLD1* specific mutation patterns T[C>A]T and T[C>T]G are enriched in the p.D402N mutant overexpressing cell line compared to the wild-type overexpressing cell line (*p* = 0.043). We comparatively analyzed the whole-exome mutational signatures between the two engineered cell lines at passage 20. *POLD1*(wt) overexpressing cells exhibited the SBS6 mutational signature, which is etiologically associated with defective DNA mismatch repair (Figure 3D), in keeping with its baseline MSI-H status and *PMS2* mutation. In contrast, the *POLD1*(p.D402N) overexpressing cells demonstrated both SBS6 and SBS20 mutational signatures. The SBS20 mutational signature is etiologically associated with concurrent *POLD1* exonuclease mutation and defective MMR deficiency (Figure 3E) and was previously identified in endometrial cancer with this co-deficiency [31,32]. Of note, a concurrent *POLD1* proofreading/MMR deficient mutational signature is marked by peak C(C>A)T and flattened C>T peaks on the 96-element single base substitution pattern plot [32], which are present in the mutational profile of our mutant cell line (Figure 3E), but not in the *POLD1*(wt) overexpressing cell line (Figure 3D). On balance, overexpression of *POLD1* p.D402N directly increased mutation counts, SNP, INDEL, and imparted a *POLD1* exonuclease deficiency specific mutational signature on the cancer genome in our engineered endometrial cancer cell line model.

## 4. Discussion

*POLD1* is the largest functional subunit of POLδ, playing a critical role in maintaining the genome with its DNA polymerase and exonuclease functions. When defective, the *POLD1* exonuclease domain mutation contributes to increased somatic mutation burdens, a predisposition to hereditary colorectal and endometrial cancer, and colonic polyposis. The exonuclease domain contains four highly conserved amino acids, DEDD, which bind to magnesium cations that catalyze phosphoryl-transfer reaction for removing misincorporated nucleotides. Due to the functional significance of the DEDD motif—particularly the negatively charged sidechains—amino acid substitution that leads to loss of the negative charge at any of the four residues should result in the loss of exonuclease activity. Despite multiple reports of germline *POLD1* variants involving changes in the DEDD motif in both colorectal and endometrial cancer patients [9,28,29,30], this mutation class is still classified as VUSs by laboratory submitters in the ClinVar database, rendering these variants clinically inactionable. Furthermore, during our ClinVar database search for this mutation class, we did not find suggestions of endometrial cancer relevance, raising the possibility of the under-recognition of this important association. We are motivated to functionally define these variants, with the overarching goal of upgrading their classification, and to change their clinical management from inactionable to actionable. 

To this end, we provided multiprong evidence to support the DNA proofreading deficiency of this mutation class. Our index patient, a germline carrier of *POLD1* p.D402N, developed endometrial cancer at a young age. Her endometrial tumor exhibited characteristics that were in keeping with a *POLD1* pathogenic phenotype, including high infiltrating lymphocytes, a high tumor mutation burden, and the presence of the SBS10d mutational signature. Her family history included endometrial cancer, which is in keeping with the *POLD1* mutation-related hereditary cancer syndrome spectrum. We further established the phenotype association with a PUBMED search for existing cases of patients with germline mutations in the DEDD domain. The additional cases that were found were associated with colorectal and endometrial cancers, which are disease-specific for *POLE*/*POLD1* exonuclease deficiency associated tumors. We further examined the functional outcomes of this mutation class with an in silico protein function prediction analysis coupled with an orthogonal in vitro biochemical assay. The result demonstrated loss of exonuclease function. Lastly, in the patient’s tumor and cell line overexpressing *POLD1* p.D402N, we found a mutational signature consistent with *POLD1* exonuclease deficiency signatures. Taken together, we believe that discordant AA substitutions in any of the DEDD amino acids may lead to the loss of exonuclease function (a mutational class phenomenon). Our conclusion is supported by a separate study which found a *POLD1* exonuclease deficiency mutational signature in endometrial cancers with mutations in *POLD1*’s DEDD motif residues D316 and E318 [32].

Our study offers several novel findings. First, we offered multiprong evidence to support the reclassification of this mutation class to likely pathogenic, and linked the *POLD1* DEDD motif mutation to human cancers. Second, these germline variants were observed in colorectal cancer registries from multiple countries (Spain, Taiwan, England), suggesting the possibility of the greater geographical distribution of these germline variants than initially realized. Third, we found an unexpected association between this mutation class and breast cancer from our literature search [28,29]. The association of *POLD1* exonuclease deficiency and breast cancer is unclear at this time. Given that breast cancer is the most common cancer in women, the co-occurrence of the *POLD1* germline variant and breast cancer development may be incidental. Prospective data collection is needed to further clarify if the *POLD1* DEDD domain altering allele increases the frequency of breast cancer development. 

Our study highlights the growing clinical problem of receiving VUS results from germline testing. With increased utilization of germline testing, we expect more patients to receive VUS results. While VUSs represent uncertainty in genomic medicine, they are clinically inactionable, and may be anxiety-inducing for patients [34]. The American College of Medical Genetics and Genomics (ACMG) has provided standards and guidelines to help classify variants [35]. However, the proposed criteria can be highly restrictive (for example, high-quality in vitro data may not always be available and disparate access to genetic testing in low resource areas may lead to the undercapture of germline variants and hereditary cancer association) and may under-classify variants that do not meet the high evidential thresholds. Currently, there is no dedicated working group to systematically reclassify VUSs. Rather, VUSs are passively reassessed through prospectively aggregated data collection deposited to the ClinVar database and expert panel, submitted from expert groups, researchers, and laboratories with disease phenotype information and interpretations [10]. The frequency and rigor of VUS reappraisal by this mechanism is unclear. For example, despite multiple reports of the pathogenic association between p.D316H, p.D316N, p.D316G, p.D402N, p.E318K variants and endometrial and colorectal cancers [9,28], these variants continue to be classified as VUSs in by laboratories based on currently available submissions in the ClinVar database. This may be due to the fact that ClinVar may not have the most up-to-date submissions and the lack of expert consensus on how to reclassify this variant class for *POLD1*. In our case, our multidisciplinary team made it possible to immediately investigate VUSs with high suspicion for pathogenicity. Our group effort (basic scientists, geneticists, oncologists, and pathologists) provided a coordinated, multiprong analysis of p.D402N variant and the functional significance of this mutation class. To address the growing problem of VUSs, their reclassification is necessary and requires integrative analysis. Our multidisciplinary model could serve as a framework for formulating local or national working groups to improve the VUS reclassification rate. To aid future research in the *POLD1* DEDD domain, we have enclosed the relevant genomic sequence information on humans, yeast, mice, and other organisms for the research community to use (Appendix A). 

An important observation in our study is that the *POLD1* exonuclease domain deficiency is associated with an immunotherapy response. Indeed, our patient showed deep, durable response to immunotherapy, despite being diagnosed with stage 4 high-grade endometrial cancer which had mixed response to neoadjuvant chemotherapy. This patient’s endometrial tumor contained genomic signatures (high TMB and GEP score) which have been shown to be predictive biomarkers for response to pembrolizumab [26]. The association between immunotherapy efficacy and exonuclease mutation in *POLD1* and *POLE* has been reported by various groups [2,36]. Among immunotherapy-treated patients, patients with *POLE*/*POLD1* mutations showed improved overall survival compared to patients without a *POLE*/*POLD1* mutation [2], suggesting *POLE*/*POLD1* mutational status as a predictive biomarker for immunotherapy efficacy. The therapy response observed in our patient, together with the tumor’s molecular profile characteristic of the *POLD1* signature, support our hypothesis that discordant AA substitution in the DEDD motif would lead to exonuclease dysfunction resulting in DNA proofreading defect and an increased level of neoantigen production, ultimately promoting anti-tumor immunity. To maximize access to personalized oncologic immunotherapy, it is crucial to identify all potentially pathogenic *POLD1* variants resulting in exonuclease deficiency. Patients carrying discordant AA substitution in the DEDD motif of *POLD1* may be functionally deficient in DNA proofreading and should be considered for immunotherapy. This can be corroborated with paired somatic tumor testing to assess tumor mutation burden status. The opportunity for precision oncologic immunotherapy provides encouragement to reclassify these variants from VUS (inactionable) to likely pathogenic (actionable). 

In summary, the findings in this study provide strong evidence for the pathogenicity of *POLD1* p.D402N and raise suspicion regarding the pathogenicity of other charge-altering mutations involving the DEDD motif of *POLD1*. Applying the ACMG [35] on the interpretation of sequence variants, we met the criteria to reclassify the p.D402N variant as possibly pathogenic for the following: in vitro functional and molecular studies are supportive of a damaging effect on the gene and the allele being at extremely low frequency in the ExAc database [27]. Other supporting evidence includes (i) multiple computational evidence supporting a deleterious effect of loss of acidic amino acid in the DEDD domain, (ii) the patient’s phenotype (early onset of endometrial cancer, high TMB and TIL), and (iii) a family history of endometrial cancer is highly specific for a disease with a single genetic etiology. Collectively, pD402N should be reclassified as likely pathogenic, so these patients may receive appropriate genetic and cancer screening follow-ups. Furthermore, the durable response of immunotherapy in our index patient is in keeping with the observation that *POLD1* is a biomarker for immunotherapy, making the reclassification of these alleles even more important for personalized oncologic care. 

## 5. Conclusions

Charge-discordant AA substitution in the DEDD motif of *POLD1* is detrimental to DNA proofreading and should be reclassified as likely pathogenic and possibly predictive of ICI sensitivity.

## Figures and Tables

**Figure 1 cancers-15-05674-f001:**
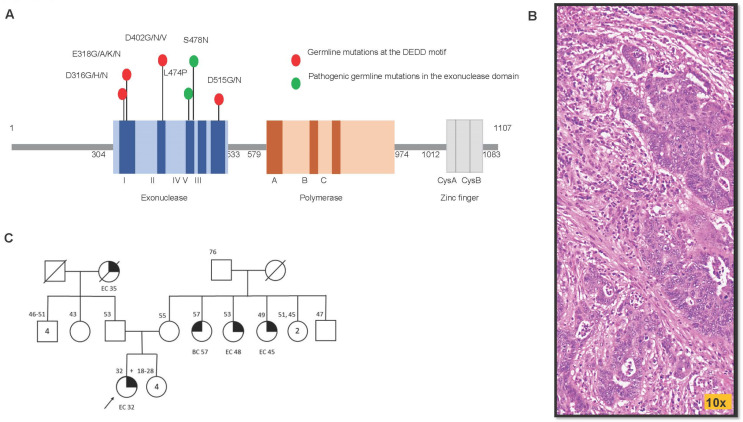
(**A**) Germline *POLD1* mutations in humans in relation to protein structure. Currently, two pathogenic variants have been identified (green circles) within the exonuclease domain; thirteen charge-altering mutations are found within the DEDD motif (red circles), all currently classified as variants of uncertain significance. (**B**) Histology (H&E) of endometrial cancer carrying the p.D402N *POLD1* mutation. Note the high amount of tumor infiltrating lymphocytes in the microenvironment, in keeping with the tumor’s high mutational status. (**C**) The family pedigree tree of the index patient with germline POLD1 p.D402N mutation. Black filled symbols indicate endometrial cancer (EC) and breast cancer (BC), respectively. Ages at diagnosis are indicated for each patient at the top of each shape. Circle and square shapes in the pedigree diagramph represent female and male gender, respectively. Number inside the shape represent number of sibling of the same gender. Black arrow denotes the index patient.

**Figure 2 cancers-15-05674-f002:**
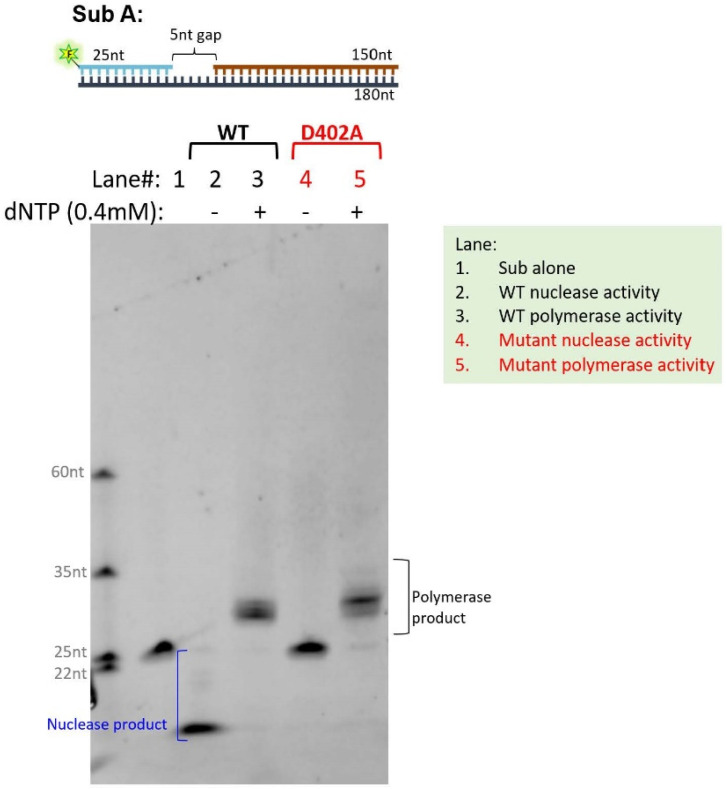
Nuclease and polymerase activities of WT and p.D402A POLD1 in the POLδ complex. The top panel shows the 5′ end FAM-labeled DNA gapped DNA duplex substrate, resembling the intermediate structure that occurs during lagging strand DNA synthesis. The bottom panel is the PAGE image of the reaction of the polymerase (with dNTPs) or 3′ exonuclease (without dNTPs) activity assay. Substrates, 3′ exonuclease or polymerase products were indicated. Abbreviations: Sub = DNA duplex substrate; WT = wild type; FAM = fluorescein; Star symbol containing F letter = fluorescein.

**Figure 3 cancers-15-05674-f003:**
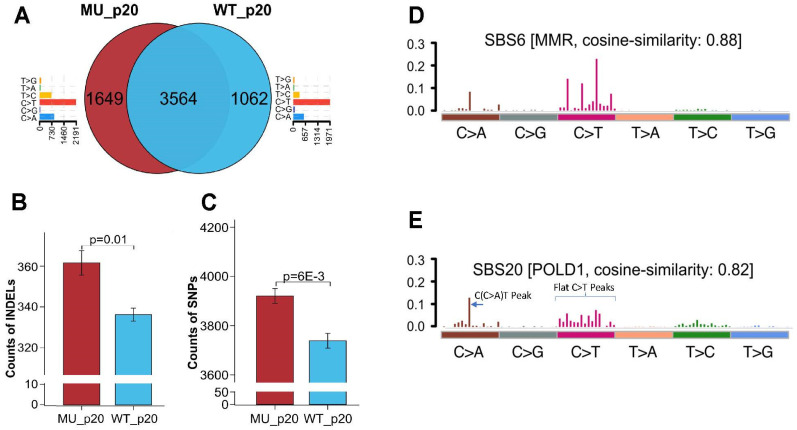
Whole exome sequencing analysis of a lentiviral vector transfected stable cell line that overexpresses either p.D402N (MU) and wild type (WT) in endometrial cancer cell line HEC1A, collected at passage 20. Venn diagram showing unique and shared mutations in MU-overexpressing (**A**) or WT-overexpressing endometrial cancer cells (**A**). The bar graph next to the Venn diagram represents the frequency of specific single base changes occurring in unshared mutations between the MUT and WT cells. Note that C>A, a base change specific for *POLD1* exonuclease deficiency, is higher in the mutant cell lines compared to the wildtype cell lines (*p* = 0.04; (**A**)). INDELs (**B**) and single nucleotide polymorphisms (SNP); (**C**) distribution in MU and WT overexpressing cell lines at passage 20. Mutational signature analysis of WT overexpressing cell detected the SBS6 mutational signature, consistent with its underlying mismatch repair (MMR) deficiency status (**D**). In MU-overexpressing cells, there is a peak frequency at the C(C>A)T single base change, with concurrent flattening of C>T frequency peaks, which are characteristic of the mutational signature pattern for concurrent *POLD1*/MMR deficiency (**E**). Three technical replicates were performed for the sequencing runs of each cell line (WT vs. MU).

**Table 1 cancers-15-05674-t001:** Genotypic and phenotypic characteristics of carriers of all reported mutations in the DEDD motif of *POLD1* (systematic review and current study).

POLD1 Variant (Male/Female)	Cancer Type	Country/Consortium	Source	Cancer Age	Colonic Polyps
p.D316G (female)	Colorectal cancer, Endometrial cancer	Spain	Bellido [28]	44, 54	Yes
p.D316G; p.V295M (female)	Endometrial cancer, Breast Cancer	Spain	Bellido [28]	57	N/A
p.D316H (female)	Breast Cancer	Spain	Bellido [28]	64	Yes
p.D316H (male)	Colorectal cancer, angiomyolipoma, mesothelioma	Spain	Bellido [28]	58	Yes
p.D316N (female)	Breast cancer	United Kingdom/CORGI	Palles [29]	52	No
p.D316N (female)	Endometrial cancer	United Kingdom/CORGI	Palles [29], Robinson [9]	54	Yes
p.E318K (not reported)	Colorectal cancer	Taiwan	Chang [30]	N/A	N/A
p.D402N (not reported)	Colorectal cancer	Taiwan	Chang [30]	N/A	N/A
p.D402N (female; current case)	Endometrial cancer	USA	Current study	32	N/A

**Table 2 cancers-15-05674-t002:** Summary of in silico protein function prediction of all 13 *POLD1* variants involving the DEDD motif.

Name/Swissport P28340	PolyPhen-2	Mutation Taster	Panther	Mutation Assessor	SIFT	MutPred2 Score *
NM_002691.4(POLD1):c.946G > C (p.Asp316His)	Probably damaging	Disease causing	Probably damaging	High functional impact	Not tolerated	0.875
NM_002691.4(POLD1):c.946G > A (p.Asp316Asn)	Probably damaging	Disease causing	Probably damaging	Medium functional impact	Not tolerated	0.744
NM_002691.4(POLD1):c.947A > G (p.Asp316Gly)	Probably damaging	Disease causing	Probably damaging	High functional impact	Not tolerated	0.882
NM_002691.4(POLD1):c.952G > C (p.Glu318Gln)	Probably damaging	Disease causing	Probably damaging	High functional impact	Not tolerated	0.782
NM_002691.4(POLD1):c.952G > A (p.Glu318Lys)	Probably damaging	Disease causing	Probably damaging	High functional impact	Not tolerated	0.885
NM_002691.4(POLD1):c.953A > G (p.Glu318Gly)	Probably damaging	Disease causing	Probably damaging	High functional impact	Not tolerated	0.898
NM_002691.4(POLD1):c.953A > C (p.Glu318Ala)	Probably damaging	Disease causing	Probably damaging	High functional impact	Not tolerated	0.878
NM_002691.4(POLD1):c.1204G > T (p.Asp402Tyr)	Probably damaging	Disease causing	Probably damaging	High functional impact	Not tolerated	0.932
NM_002691.4(POLD1):c.1204G > A (p.Asp402Asn)	Probably damaging	Disease causing	Probably damaging	High functional impact	Not tolerated	0.884
NM_002691.4(POLD1):c.1205A > T (p.Asp402Val)	Probably damaging	Disease causing	Probably damaging	High functional impact	Not tolerated	0.924
NM_002691.4(POLD1):c.1205A > G (p.Asp402Gly)	Probably damaging	Disease causing	Probably damaging	High functional impact	Not tolerated	0.926
NM_002691.4(POLD1):c.1543G > A (p.Asp515Asn)	Probably damaging	Disease causing	Probably damaging	High functional impact	Not tolerated	0.779
NM_002691.4(POLD1):c.1544A > G (p.Asp515Gly)	Probably damaging	Disease causing	Probably damaging	High functional impact	Not tolerated	0.877

* MutPred2 Score threshold greater than 0.5 supports pathogenicity.

## Data Availability

The data produced in this study can be obtained from the corresponding author upon reasonable request.

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
