# Peer review of "POLD1* DEDD Motif Mutation Confers Hypermutation in Endometrial Cancer and Durable Response to Pembrolizumab"

_cancers, 2023, doi:10.3390/cancers15235674_

Round 1

Reviewer 1 Report

Comments and Suggestions for Authors

Comments on the manuscript cancers-2736324 is an interesting study in which the researchers describe a clinical case of an endometrial cancer patient who is a carrier in a heterozygous state for the POLD1 variant (p.D402N), and performs bioinformatic analysis of the implications of the p.D402N variant through PRISMA, orthogonal biochemical confirmation, and whole exome and RNA sequencing analysis of the patient's tumor and engineered cell lines.

However, a number of comments are listed below

1. In general, it is necessary to correct multiple impressions throughout the text, such as: including electronic addresses of the programs used or databases and the date of consultation. Some abbreviations found in the text are not previously defined, in silico, in vivo put in italics, references do not have an ordered citation in the text, signs that refer to degrees Celsius are missing, among others, some subtitles refer to them with a black dot is suggested to refer to the bold subtitles. Homogenize the list of references according to the instructions for the author

Figure 1B be more specific and indicate with a mark what you want to highlight in the figure, figure 1C refers to EC and BC and in the figure caption EnCa, just as breast cancer is confusing, it is suggested to homogenize.

In Table 2, integrate the asterisk in the appropriate column.

Improve the resolution of the photograph included in Figure 1. As mentioned in their results and discussion, it is suggested to design a graphic scheme that explains the possible changes generated by the variants in the POLD1 gene and the molecular pathways where they can interact and how this may influence the response to pembrolizumab therapy

Author Response

  1. In general, it is necessary to correct multiple impressions throughout the text, such as: including electronic addresses of the programs used or databases and the date of consultation. Some abbreviations found in the text are not previously defined, in silico, in vivo put in italics, references do not have an ordered citation in the text, signs that refer to degrees Celsius are missing, among others, some subtitles refer to them with a black dot is suggested to refer to the bold subtitles. Homogenize the list of references according to the instructions for the author.

Thank you for the feedback. We have made all the requested changes. For example, we have included the electronic addresses of the in silico programs used as Supplemental Table 2 (unfortunately, we did not capture the date of when we ran the program, so we included the approximate time span for running the algorithms online, which was from 1/2022-1/2023) in the Methods Section under the subheading “In silico analysis predicting functional effect of DEDD domain alteration.” All gene names (e.g. POLD1), in silico, and in vitro, have all been italicized. We have gone through the references in the text, which are now ordered and use the recommended formatting style. Degree Celsius has been clarified. We have corrected all the black dot symbols and replaced with the appropriate symbol.

  1. Figure 1B be more specific and indicate with a mark what you want to highlight in the figure, figure 1C refers to EC and BC and in the figure caption EnCa, just as breast cancer is confusing, it is suggested to homogenize.

Thank you for the feedback. For figure 1B, unfortunately, due to the size and format type of the figure we are unable to superimpose a mark directly on the image, which will inadvertently cover the image. For figure 1 C, we have corrected the caption to unify the abbreviations.

  1. In Table 2, integrate the asterisk in the appropriate column.

Thank you, we have placed the asterisk in the upper right hand corner of the table (MutPred2 Score).

  1. Improve the resolution of the photograph included in Figure 1.

Thank you for your feedback. Unfortunately, the image quality is dependent, in part, to our microscope imaging acquisition equipment. To our best effort, this is the image quality captured by our equipment.

  1. As mentioned in their results and discussion, it is suggested to design a graphic scheme that explains the possible changes generated by the variants in the POLD1 gene and the molecular pathways where they can interact and how this may influence the response to pembrolizumab therapy.

Thank you for your feedback, due to the number of images that are already included in the manuscript, we feel that adding a graphic scheme may increase the page length. To convey our conclusion succinctly, we included a Simple Summary at the beginning of the manuscript to help convey our manuscript’s synopsis.

Reviewer 2 Report

Comments and Suggestions for Authors

The manuscript entitled POLD1 DEDD Motif Mutation Confers Hypermutation in Endometrial Cancer and Durable Response to Pembrolizumab, is well organized. The authors performed several functional assays in order to demonstrate that POLD1 p.D402N variant must be re classified us likely pathogenic. 

Personal, I had a similar family, but the variant was a deletion and is classified us likely pathogenic. 

Minor aspects:

Check again all the clinical significance and the evaluation data from supp. material- al the data mentioned by the authors has been changed 
